# Association between Long-Term Ambient PM2.5 Exposure and under-5 Mortality: A Scoping Review

**DOI:** 10.3390/ijerph20043270

**Published:** 2023-02-13

**Authors:** Wahida Musarrat Anita, Kayo Ueda, Athicha Uttajug, Xerxes Tesoro Seposo, Hirohisa Takano

**Affiliations:** 1Graduate School of Global Environmental Studies (GSGES), Kyoto University, Kyoto 615-8540, Japan; 2Department of Hygiene, Graduate School of Medicine, Hokkaido University, Hokkaido 060-8638, Japan; 3Graduate School of Engineering, Kyoto University, Kyoto 615-8540, Japan

**Keywords:** ambient PM2.5, under-5 mortality, post-birth exposure, epidemiological studies, review

## Abstract

Studies have established a link between exposure to fine particles (PM2.5) and mortality in infants and children. However, few studies have explored the association between post-birth exposure to PM2.5 and under-5 mortality. We conducted a scoping review to identify relevant epidemiological evidence on the association between post-birth ambient PM2.5 exposure and under-5 mortality. We searched PubMed and Web of Science for articles published between 1970 and the end of January 2022 that explicitly linked ambient PM2.5 and under-5 mortality by considering the study area, study design, exposure window, and child age. Information was extracted on the study characteristics, exposure assessment and duration, outcomes, and effect estimates/findings. Ultimately, 13 studies on infant and child mortality were selected. Only four studies measured the effect of post-birth exposure to PM2.5 on under-5 mortality. Only one cohort study mentioned a positive association between post-birth ambient PM2.5 exposure and under-5 mortality. The results of this scoping review highlight the need for extensive research in this field, given that long-term exposure to ambient PM2.5 is a major global health risk and child mortality remains high in some countries.

## 1. Introduction

Child mortality, or under-5 mortality, refers to the probability of a child dying between birth and age 5 and is an important indicator of a country’s overall state of child healthcare and socioeconomic development [1]. Child mortality has declined substantially, from 12.5 million in 1990 to 5.2 million in 2019, with worldwide efforts to reach Sustainable Development Goals (SDGs) on child survival [2]. However, child mortality rates remain high in low- and middle-income countries (LMICs), particularly in Sub-Saharan Africa and South Asia [3]. In these countries, the leading causes of child mortality include lower respiratory diseases and neonatal preterm birth complications [4], both of which have been linked to ambient air pollution. According to the World Health Organization (WHO), air pollution is one of the leading risks to child health [5,6]. In 2016, more than 90% of the world’s children, particularly the 630 million in LMICs, were exposed to higher levels of ambient particles with diameters less than 2.5 μm (PM2.5) than those considered healthy by WHO guidelines [6].

Evidence suggests that children are particularly vulnerable to air pollution, primarily due to their smaller airways, immature respiratory and immune systems, higher respiratory rate, and relatively high absorption of pollutants [7,8]. Previous epidemiological studies reported an association between exposure to PM2.5 and various health outcomes for children, such as respiratory infections [9,10], asthma [11,12], preterm birth [13,14], low birth weight [15], growth impairment [16], and neurodevelopmental [17,18] and metabolic [19] diseases, all of which can increase mortality risk.

Exposure to particulate matter both before and after birth can increase the risk of infant and child mortality. Several studies estimated the risk of infant/child mortality from exposure to PM2.5 [15,20,21], and some also estimated the risk for different exposure windows to explore the sensitive time period [15,22,23]. These studies revealed that the association between exposure to ambient PM2.5 and infant mortality remains inconclusive. The association between post-birth ambient PM2.5 and under-5 mortality is also unclear, as the impact of ambient air pollution on child mortality has not been quantified until recently. Many of these studies evaluated the association between indoor air pollution exposure and child mortality [24,25,26], focusing less on the association between long-term ambient PM2.5 and under-5 mortality. Moreover, determinants of neonatal mortality and under-5 mortality do not always overlap; this is because, with neonates, maternal exposure to air pollution during the gestational period is often considered a key contributor to severe birth outcomes leading to neonatal death [27,28,29].

Under-5 mortality is a fundamental indicator of public health. Quantitative targets for reducing under-5 mortality have been a part of several global agendas in the past and also represent a critical objective of Sustainable Development Goal (SDG) 3, which aims to reduce under-5 mortality to at least 25 deaths per 1000 live births [30]. The identification of potential avoidable risk factors for under-5 mortality is important if countries hope to achieve the SDG targets of clean air and child health. To date, however, research on the association between long-term exposure (post-birth) to ambient PM2.5 and under-5 mortality has been limited. Against this backdrop, the present scoping review aimed to identify relevant epidemiological evidence on the association between long-term (post-birth) ambient PM2.5 exposure and under-5 mortality, and how these are impacted by the study region and exposure window.

## 2. Materials and Methods

### 2.1. Data Source and Article Search

A scoping review [31] was conducted to explore epidemiological studies on the impacts of long-term exposure to ambient PM2.5 on children. We searched for articles reporting on the effects of long-term ambient PM2.5 exposure on under-5 mortality using two databases: Medline via PubMed and Web of Science. The target publication period was between 1 January 1970 and 31 January 2022. Two independent reviewers conducted the article search (WMA and KU).

We followed the guidelines of the PRISMA Extension for Scoping Reviews (PRISMA-ScR) [32]. A search strategy was developed and a literature search was conducted by combining search terms related to exposure (“ambient particulate matter” OR “outdoor air pollution” OR “outdoor PM2.5” OR “air pollutant”), health outcomes (“mortality” OR “death *”), target population (“child * OR “infant *” OR “under five”), time period of exposure (“long term”), and study design (“cohort studies” OR “cohort *” OR “epidemiological” OR “longitudinal”). We used Rayyan QCRI software to screen for articles. Rayyan was developed as a free mobile and web-based tool with the objective of facilitating the abstract/title screening stage of systematic reviews and other knowledge synthesis processes with an intelligent and more user-friendly interface [33]. Rayyan has been used to conduct reviews in different health-related research areas [34], including neurological research [35], given its ability to connect many collaborators and the intuitiveness of the system. We also searched the bibliographies of the identified articles to identify other relevant articles on the topic.

### 2.2. Selection of Articles

After the initial search, potential duplicates were removed before starting the screening process. Two independent reviewers (WMA and KU) initially screened the articles based on titles and abstracts. Full-text screening was then conducted independently if deemed necessary by the reviewers. We included original peer-reviewed articles in English-language journals with any study design as well as ambient PM2.5 from any source, in any location, and for any size of the study population. We excluded articles on indoor/household air pollution and short-term/daily exposure, those which did not relate to child mortality (e.g., subjects are all adults) or ambient PM2.5 exposure, and those which were not original research articles (e.g., reviews, protocols for future research projects, letters, and background documents). Some articles contained more than one exclusion criterion, and we resolved this issue by categorizing the article under the most relevant keyword according to the content of the article.

During the full-text screening stage, both reviewers excluded a few more studies that did not match the study objective and inclusion criteria. We excluded studies that only emphasized gestational exposure and pregnancy outcomes, such as stillbirth, low birth weight, and miscarriage, because the focus of those studies was limited to maternal exposure, whereas our focus was on under-5 mortality attributable to lifetime or post-birth exposure, which differs from birth outcomes. We also excluded articles that focused only on neonatal deaths because those deaths are more likely attributable to maternal exposure or other birth complications. All relevant articles were selected irrespective of the source of PM2.5, the socioeconomic context of the subjects, and urban/rural settings.

### 2.3. Data Extraction and Analysis

Two reviewers extracted information on the author’s name, publication year, study population, study location, study design, type of pollutant, duration of exposure, exposure measurement method, effect size (or findings), and health outcomes. Some of the studies analyzed the effect of prenatal or maternal exposure and lifetime postnatal exposure together to find the impact on infant mortality. Those studies defined the prenatal period as that spanning the beginning of the 1st trimester through the 3rd trimester of the gestational period. In the present study, we defined infant mortality as death within the first year of life (age, 0–364 days) and under-5 mortality as the death of a child before they reached 5 years of age. Infant mortality was classified into two categories: neonatal mortality (age, 0–28 days) and post-neonatal mortality (age, 28–364 days). Effect estimates are presented as percent changes in excess risk with 95% confidence intervals (95% CIs) per 1 µg/m^3^ increase in ambient PM2.5. We defined long-term exposure as the post-birth lifetime exposure of the child from the day of their birth until the day of the outcome.

## 3. Results

Figure 1 is a flowchart summarizing the article selection process. The primary search yielded 552 articles, of which 293 were found in PubMed and 259 were from Web of Science. Duplicate entries (*n* = 58) were removed before starting the screening process.

After screening by title and abstract, 400 articles were removed, and the remaining 94 articles were included for full-text reading and assessed for eligibility. Among the 94 articles, 81 were excluded because they focused only on pregnancy outcomes (*n* = 39), only presented results for adult populations (*n* = 19), used different outcomes, such as pregnancy outcomes, or targeted other diseases, such as anemia, cancer, and growth impairment (*n* = 11). Non-English articles (*n* = 7) and articles for which the full text was not available (*n* = 2) were also excluded. We also excluded articles that used a definition of “child” that encompassed those aged >5 years (*n* = 3). After the screen, a total of 13 articles remained for subsequent analyses (Table 1).

### 3.1. Study Characteristics

Table 2 shows the general characteristics of the 13 selected articles. Of these, three targeted more than one country, e.g., LMICs in different parts of the world [38,39,45]. Two studies were conducted in India [36,46], three were in the United States [22,41,42], three were in Africa [37,44,47], and two were in South Korea [40,43].

Four of the selected studies were cohort studies [22,36,37,40], four were cross-sectional studies [44,45,46,47], two were case–control studies [42,43], and two were ecological studies [38,39]. The design of the study by Khadka and colleagues [41] was unclear from the text. However, since the study mentioned using “cohort-linked” population data and measured individual-level health outcomes, we categorized it as a cohort study, bringing the total number of cohort studies to five.

### 3.2. Outcomes

Outcomes of child mortality were categorized by child age and cause of death. Four studies [36,37,38,39] focused on under-5 mortality. The study by Goyal and colleagues examined associations between exposure in the first year of the child’s life and infant mortality, neonatal mortality, and post-neonatal infant mortality [45]. Studies from the United States [22,42] and South Korea [40,43] focused on post-neonatal mortality, which was classified as infant mortality. The study from India by deSouza and colleagues [46] focused on neonatal, post-neonatal, and infant mortality. Two Studies from Africa [44,47] and another study from the United States [41] focused on infant mortality. While all studies examined the association between PM2.5 and all-cause mortality, four studies [22,37,40,42] focused on respiratory causes; of these, three [22,40,42] also presented results for sudden infant death syndrome (SIDS).

### 3.3. Exposure Assessments

Exposure assessments are summarized in Table 3. Three studies used measurements from a ground-based monitoring station [40,42,44], while the remaining studies used either satellite-retrieved data [22,38,39,44,45,46] and/or data from different models [36,41] (such as the Community Multiscale Air Quality model (CMAQ) [43], atmospheric chemistry model of the Goddard Earth Observing System [47] (GEOS-Chem), and a combination of atmospheric models (CMAQ) with ground-based measurements) [41]. The study by Liao et al. developed a machine learning random forest (RF) model to predict monthly ambient PM2.5 concentrations [36].

Four studies [38,39,45,47] reported on diverse types and sources of PM2.5 and constituents of ambient PM2.5 estimated from atmospheric modeling. PM2.5 constituents included natural (mostly desert dust and sea salt) and anthropogenic (industrial emissions, biomass burning, and agriculture) sources, biomass/dust mixtures, anthropogenic/dust mixtures, and biomass/anthropogenic mixtures [38,39,45]. The study by Bachwenkizi et al. analyzed PM2.5 mass concentrations and its six constituents, i.e., organic matter, black carbon, sulfate, nitrate, ammonium, and soil dust [47].

Three studies measured the effects of lifetime (post-birth) exposure to PM2.5 on under-5 mortality [37,38,39]. Three studies measured the effects of post-birth exposure on infant mortality [22,42,47]. The remaining studies assessed the effects of both in utero and post-birth exposure on infant/child mortality. Khadka et al. examined the relationships of prenatal and post-birth exposure with infant death and estimated direct paths from the two exposures to infant death as well as indirect paths from prenatal exposure to the outcomes (preterm birth and low birth weight) [41].

### 3.4. Adjusted Covariates

All selected studies adjusted their results for different covariates based on key determinants of infant and child mortalities in the respective countries. These included covariates at the maternal level (age, education, marital status, smoking habit, race, and ethnicity), child level (birth weight, sex, length of gestation period, time of birth, and parity), household level (income, wealth, employment, fuel use, improved sanitation, and access to safe drinking water), and country level (seasonality, humidity, temperature, and rainfall), as well as other socioeconomic variables, such as the place of residence, wealth index, and socioeconomic status. Some studies also included specific covariates in the analyses, such as records of diarrhea and vaccination [36] and national healthcare service expenditures [47].

### 3.5. Effect Estimates

#### 3.5.1. Post-Birth Exposure to PM2.5 and under-5 Mortality

Four studies discussed the effects of post-birth or lifetime exposure to ambient PM2.5 on under-5 mortality. In a study from India, Liao and colleagues used the Cox proportional hazard regression model and reported a 1.30% [95% (CI): 0.10%, 2.60%] increased risk of under-5 mortality for post-birth lifetime exposure to ambient PM2.5 and a 2.30% [95% (CI): 0.8%, 3.8%] elevated risk of in utero exposure for a 10 µg/m^3^ increase in ambient PM2.5 [36]. A study from Kenya using logistic and Poisson regression models reported that exposure to high levels of air pollution (PM2.5 ≥ 25 µg/m^3^) was associated with a 22.00% [95% (CI): 8.00%, 39.00%] increase in the risk of mortality [37]. Meanwhile, Lien et al. reported a significant increase in the risk of mortality for biomass PM2.5 exposure [29.00%; 95% (CI): 13.11%, 47.13%] and a moderate increase for anthropogenic PM2.5 exposure [12.00%; 95% (CI): 1.09%, 24.10%] by using a generalized additive mixed-effects model [38]. Owili et al. reported an 8.90% [95% (CI): 4.11%, 13.91%] increase in the risk of under-5 mortality for biomass PM2.5 and marginal or no associations with PM2.5 from other sources using the generalized linear mixed-effects model [39].

#### 3.5.2. Exposure to PM2.5 and Infant Mortality

The association between ambient PM2.5 exposure (both in utero and post-birth) and infant mortality was reported by eight studies. The results for in utero exposure were dependent on the trimester, cause of death, study design, and postnatal lifespan of the infant. Findings on the effects of in utero exposure in different trimesters on infant mortality were somewhat contradictory. Son et al., using a Cox proportional hazards model, reported a 14.50% [95% (CI): 6.70%, 22.80%] increased risk of infant mortality due to gestational exposure [40], and deSouza and colleagues, using fixed-effects regression, reported an increase of 1.60% due to exposure only in the third trimester [46]. Other studies found no association between gestational or in utero exposure and any stage of infant mortality [22,45]. For postnatal or lifetime exposure, Heft-Neal et al., using a fixed-effects regression model, reported a 9.00% [95% (CI): 4.00%, 14.00%)] increase in infant mortality from all causes for a 10 µg/m^3^ change in ambient PM2.5 [44], and Woodruff et al. reported a 7.00% [95% (CI): −7.33%, 23.55%] increase in all-cause mortality and a 113.00% [95% (CI): −12.01%, 305.04%] increase in deaths related to respiratory causes using a conditional logistic regression model [42]. For post-birth exposure, Son et al. used an extended Cox proportional hazards model and reported a 112.24% [95% (CI): 71.72%, 162.32%] increase in infant deaths in the United States [22], while Bachwenkizi and colleagues reported a 3.00% [95% (CI): 0.54, 5.51%] increase in infant deaths using multivariable logistic regression analysis [47]. In contrast, Khadka and colleagues reported no significant increase in infant mortality due to post-birth exposure to ambient PM2.5; notably, their study did not provide information about the effect size [41].

## 4. Discussion

The present scoping review identified 13 epidemiological studies that examined the association between long-term post-birth ambient PM2.5 exposure and under-5 mortality. Among these studies, four focused on the relationship between under-5 mortality and long-term exposure to PM2.5, whereas the remaining studies focused on infant mortality. The results of this scoping review reveal the relative lack of research on post-birth PM2.5 exposure and under-5 mortality and highlight the importance of engaging in further research on this topic.

Among the selected studies, study designs differed mainly in studies targeting LMICs. The outcome for the selected studies was infant and child death attributable to long-term exposure to ambient PM2.5. The majority of studies suggested a positive association between lifetime exposure to ambient particulate matter (PM2.5) and infant and child mortality, although there was variation in the effect size among studies, and even within the same study in some cases. Studies differed in their design, sample size, exposure measurement method, and many other aspects, which likely led to the observed variation. There were also specific differences between studies, such as the types and sources of PM2.5, which led to different effect estimates. For instance, Owili et al. reported that biomass PM2.5 was significantly associated with an increased risk of overall global under-5 mortality, whereas anthropogenic PM2.5 was not associated with under-5 mortality, although higher concentrations of anthropogenic PM2.5 were found in different parts of the study area [39]. Meanwhile, Lien et al. reported that both anthropogenic and biomass PM2.5 exposure elevated the risk of under-5 mortality overall in Asia [38]. For younger children, PM2.5 derived from carbonaceous and anthropogenic sources was associated with an increased risk of mortality [45]. The variation in results among the selected studies may be due to the different global frequency distributions of various PM2.5 types in different regions and time periods. In addition, the large study area and the mixture of PM2.5 sources may have caused some overestimation or underestimation of results, as well as some inconsistencies. Therefore, it was difficult to gain a clear understanding of the impact of PM2.5 and its constituents on under-5 mortality.

Our findings were also inconclusive for the studies that evaluated the mass concentration of PM2.5 as the environmental exposure instead of the discriminated PM2.5. According to the measurement scale of the association, a study conducted in Kenya reported an increased risk of all-cause mortality in under-5 children who live in areas with high levels of air pollution (PM2.5 > 25 µg/m^3^) [37]. Risk estimates from such studies are difficult to compare with studies that reported results as risk per unit increase in PM2.5 levels. Other included studies on PM2.5 and infant mortality that analyzed PM2.5 mass concentration data reported varying results. Most of these reported positive associations between ambient PM2.5 exposure and infant mortality up to age 1 and did not address mortality in children older than 1 year attributable to PM2.5 exposure. From these results, it was not possible to infer how post-birth exposure to PM2.5 influenced under-5 mortality.

The exposure time frame or exposure window is one of the most important characteristics of air pollution health research. We noticed that differences in the critical exposure window led to different results for infant mortality in different trimesters in utero [40,43,46] and lifetime exposure [22,40,46]. In some cases, different results were reported within the same study for post-birth lifetime exposure depending on the source and components of PM2.5 [38,39,47]. This heterogeneity made it difficult to summarize the findings or draw inferences regarding the association between post-birth lifetime exposure to ambient PM2.5 and child mortality. These discrepancies could be explained by the complexity of maternal health, as well as by our inadequate understanding of the biological mechanisms through which exposure to PM2.5 leads to infant mortality [43]. Another potential reason is reporting bias, as many of the selected studies relied on mother-reported cross-sectional survey data rather than hospital records. Other plausible causes of the observed inconsistencies include the different exposure assessment methods used, as well as not controlling for potential confounders at the individual level, particularly for studies with an ecological design.

The most recent cohort study of under-5 mortality in India conducted by Liao et al. was the only directly on-topic study we identified in the entire review process. Although the study estimated child survival for both in utero exposure and post-birth lifetime exposure, the results were presented separately, and associations were estimated using both single-pollutant and two-pollutant models [36]. The study by Liao et al. was particularly important for understanding the association between lifetime exposure to ambient PM2.5 and under-5 child mortality in LMICs under high pollution scenarios.

The present scoping review made evident the research gaps in this particular field and highlighted the need for more research on the impact of long-term ambient PM2.5 on under-5 mortality. Most of the selected studies were from LMICs, and we did not identify any that examined the association between under-5 mortality and post-birth ambient PM2.5 exposure from high-income countries. This may have been due to the relatively low concentrations of PM2.5 in high-income countries compared with highly polluted regions such as South Asia, as well as the comparatively low under-5 mortality rate resulting from advances in healthcare, access to clean water and sanitation, and improvements in the overall quality of life. However, other studies demonstrated that socioeconomic, racial, regional, and other disparities in maternal health and infant/child mortality prevail in high-income countries [48], and that premature mortality can increase even at low concentrations of PM2.5 [49,50]. Further, more extensive research on PM2.5-related mortality is required for high-income countries as well. The present study also revealed the scarcity of relevant cohort studies from developing countries. Nevertheless, in order to understand the potential risk of under-5 mortality, more research in this field is required, particularly given the growing concerns around the world regarding the fatal impact of ambient air pollution exposure on child mortality, especially in South Asia and Sub-Saharan Africa. In South Asian countries, ambient PM2.5 concentrations are manifold higher than the WHO permissible limit and their own air quality standards; moreover, in some of the countries, under-5 mortality is still very high [51]. A recent report from 2020 targeting South Asia reported a sharp rise in infant and child mortality compared to the previous year, as many basic healthcare facilities were interrupted due to the COVID-19 pandemic [3]. More comprehensive studies are needed to better understand the risk of air pollution-attributable under-5 mortality in order to take appropriate measures.

### Strengths and Limitations

One strength of this scoping review is that, to our knowledge, it is the first review to address the impact of long-term (after birth) ambient PM2.5 exposure on under-5 mortality by considering exposure windows. While several reviews have considered both long- and short-term exposure, exposure to different air pollutants, and both infant and child mortality, most of these reviews also considered gestational or in utero exposure and not only postnatal lifetime exposure. Therefore, the present study serves as a good starting point for engaging in further in-depth research on the association between exposure to ambient PM2.5 and under-5 mortality for both high- and low-pollution areas.

The present study also has several limitations worth noting. First, the number of selected studies that examined the effect of long-term exposure to PM2.5 on under-5 children was low for both developed and developing countries, and the studies also had differing designs. Thus, we could not perform quantitative analyses. Second, the results were heterogeneous, even within the same study in some cases. As mentioned earlier, ecological studies on long-term ambient PM2.5 and under-5 mortality in global settings used different types of PM2.5 when discussing effect sizes. Thus, it was difficult to understand the measure of the association between long-term exposure to ambient PM2.5 and under-5 mortality. This is because when we measure ambient PM2.5-attributable health risks in any age group of a population, it is difficult to differentiate exposures and outcomes based on types of PM2.5. Third, we only selected studies that were published in English and in international journals. Some countries may have epidemiological studies on this topic published in their local language in local journals that we did not have access to.

## 5. Conclusions

The present scoping review highlights the lack of evidence in the literature on the effects of long-term exposure to ambient PM2.5 on child mortality. More research is required in this field since ambient air pollution is a major public health risk and, in some parts of the world, child mortality is still very high. The global community is committed to achieving SDG targets by 2030, and Goal 3 of this global blueprint aims to reduce both under-5 mortality and the number of deaths and illnesses from environmental pollution. The tracking record of SDG progress shows that many countries—especially LMICs—will not be able to meet their targets by 2030 for a number of reasons, research and data gaps being two of the most glaring to date. Therefore, in order to implement necessary policy measures to reduce excess mortality and achieve SDG targets on time, it is extremely important for governments, academia, and all other stakeholders to understand the relative risk of excess death among under-5 children due to exposure to long-term ambient PM2.5 pollution. Environmental health research can focus on obtaining more epidemiological evidence to understand the current association between PM2.5 and under-5 mortality, and how population dynamics and interventions such as improved healthcare facilities and air pollution control measures can change the burden of mortality. The exposure–response relationship ascertained from these studies could be applied to projection studies for under-5 mortality in the coming years, especially amidst growing concerns about the impact of global climate change.

## Figures and Tables

**Figure 1 ijerph-20-03270-f001:**
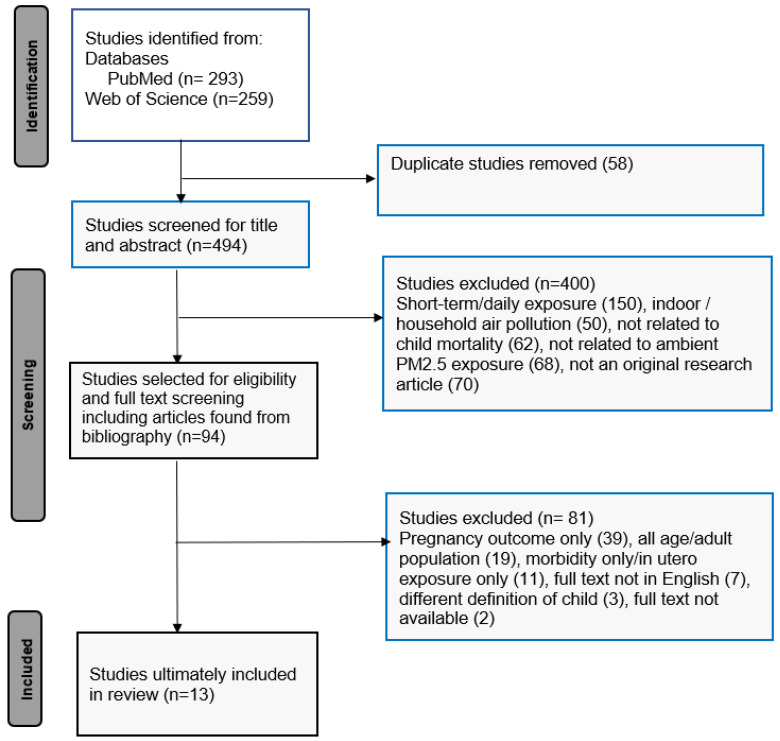
Flowchart of selection process.

**Table 1 ijerph-20-03270-t001:** Data extracted from the 13 selected articles.

Author and Year	Study Area, Country, and Study Period	Study Design and Sample Size	Health Outcome	Cause of Death	Adjusted Covariates	Statistical Model Used	Measures of Association and Selected Results of Percent Change in Excess Risk (95% CI) per 1 µg/m^3^ Increase in Particles (Unless Otherwise Stated)
					Under-5 mortality		
Liao et al., 2022 [36]	India, 2006–2016	Cohort 259,627 live-birth children born 5 years preceding the survey from 175,865 women	Neonatal mortality, infant mortality, and child mortality	All-cause	Sex, birth month/year, birth order, location (hospital or not), age of mother, height, marital status, smoking, education level, passive/active smoking, location of house, area, cooking fuel, toilet facilities, wealth index, and monthly rainfall and temperature	Cox proportional hazards regression model	HR 0.13% [95% CI: 0.01%, 0.26%] for post-delivery lifetime exposure; 0.23% [95% CI: 0.09%, 0.38%] for in utero exposure
Egondi et al., 2018 [37]	Nairobi, Kenya, 2003–2013	Cohort 21,641 children under age 5	Under-5 mortality and morbidity	All-cause and respiratory	Sex, age, and socioeconomic status	Logistic and Poisson regression models	IRR Effect size was mentioned considering the exposure level binary (PM2.5 < 25 ≥ µg/m^3^): 22.00% [95%CI: 8.00%, 39.00%]; for all-cause: 12.00% [95%CI: −12.0%, 42%).
Lien et al., 2019 [38]	45 countries in Asia by clustering in 5 regions, 2000–2015	Ecological	Under-5 and maternal mortality	All-cause and respiratory	Country, year, total population, urban population, female population, employed population, HIV/AIDS-related death, TB death, undernourished population, and temperature	Generalized additive mixed-effects model	IRR 29.00% [95% (CI): 13.11%, 47.13%] for biomass PM2.5; 12.00% [95% (CI): 1.09%, 24.10%] for anthropogenic PM2.5.
Owili et al., 2020 [39]	Global, 2000–2015	Ecological	Under-5 and maternal mortality	All-cause	Total number of undernourished, anemic pregnant women, tuberculosis cases, AIDS-related deaths, employed females, population in urban areas, year, country, and annual mean temperature	Generalized linear mixed-effects model	RR 8.90% [95% (CI): 4.11%, 13.91%] increase for biomass PM2.5 and 9.50% [95% (CI): −0.03%, 20.23%] for dust PM2.5
					Infant mortality		
Son et al., 2011 [40]	Seoul, South Korea, 2004–2007	Cohort 359,459 infants	Post-neonatal infant mortality	All-cause, respiratory, and SIDS	Sex, gestational length, season of birth, maternal age and educational level, and heat index	Cox proportional hazards model	HR 14.45% [95% (CI): 6.68%, 22.79%] increase in all-cause mortality for gestational exposure
Son et al., 2017 [22]	Massachusetts, USA, 2001–2007	Cohort 465,682 infants	Post-neonatal infant mortality	All-cause, respiratory, and SIDS	Sex, birth weight, length of gestation, mother’s age, educational level, race/ethnicity, marital status, parity, and season of birth	Cox proportional hazards model	HR 112.24% [95% (CI): 71.72%, 162.32%] increase in all-cause infant mortality for post-birth lifetime exposure
Khadka et al., 2021 [41]	USA, 2011–2013	Cohort 10,017,357 live births and 58,913 infant deaths	Infant mortality	All-cause	Individual level (age of parents, race, level of education, maternal smoking, marital status, and parity), county level (average temperature, precipitation, and unemployment), pregnancy covariates (method of payment for delivery, child sex, and multiple births), racial composition, poverty rate, median housing value, and number of physicians per 1000 persons	-	No effect size was mentioned. Prenatal PM2.5 exposure was positively associated with infant death in all trimesters; most significant relationship in third trimester; relationship between post-birth PM2.5 exposure and infant mortality was positive but less precisely estimated
Woodruff et al., 2006 [42]	California, USA, 1999–2001	Matched case–control 3877 infants	Post-neonatal infant mortality	All-cause, respiratory, and SIDS	Maternal race, education, age, marital status, and parity; confounder: birth weight	Conditional logistic regression model	OR 7.00% [95% (CI): −7.33%, 23.55%] for all-cause mortality; 113.00% [95% (CI): −12.01%, 305.04%] for respiratory mortality
Jung et al., 2020 [43]	South Korea, 2010–2015	Case–control 2,501,836 infants	Post-neonatal infant mortality	All-cause and respiratory	Maternal education, season of birth, birth weight (kg), gestational age (weeks), and region	Conditional logistic regression model	OR 2.58% [95% (CI): 0.70%, 4.50%] for infant mortality from gestational exposure 1.75% [95% (CI): 0.27%, 3.26%] and 1.92% [95% (CI): 0.0.42%, 3.45%] for exposure in 1st and 2nd trimesters, respectively
Heft-Neal et al., 2018 [44]	30 countries in Sub-Saharan Africa, 2000–2015	Cross-sectional	Infant mortality	All-cause	Child sex, birth order, age of mother, maternal education, type of cooking fuel, asset-based wealth index, temperature, and seasonal variability	Fixed-effects regression model	RR 0.87% [95% (CI): 0.40%, 1.33%] for infant mortality
Goyal et al., 2019 [45]	43 low- and middle-income countries, 1998–2014	Cross-sectional 534,476 children	Neonatal and post-neonatal mortality and infant mortality	All-cause and respiratory	Child-level (birth order, sex, multiple births, and birth interval), parent-level (age, smoking habit, and education of both parents), and household-level (location, cooking fuel, drinking water, improved sanitation, and wealth quantiles) characteristics	Multivariate logistic regression model	Odds Ratio (OR) 22.0% [95% (CI): 10.62%, 34.54%] increase in neonatal mortality and 13% [95% (CI): 3.90%, 22.89%] increase in infant mortality for anthropogenic PM2.5
deSouza et al., 2021 [46]	India, 2015–2016	Cross-sectional 259,627 children	Neonatal, post-neonatal, and infant mortality	All-cause	Child-level (sex and birth order), mother-level (age of marriage, education, age of giving birth, and smoking habit), and household-level (location, sanitation, fuel use, and safe drinking water access) covariates, including seasonality and long-term trend	Fixed-effects regression model	OR 0.16% [95% (CI): 0.03%, 0.30%] increase in neonatal mortality for exposure in 3rd trimester
Bachwenkizi et al., 2021 [47]	15 countries in Africa, 2005–2015	Cross-sectional 602,863 participants	Infant mortality	All-cause	Mother-level (age, education, and smoking habit), child-level (sex, birth order, vaccination record, and diarrhea), household-level (toilet facilities, drinking water, cooking fuel, and wealth index), and country-level (anemia, health expenditure, child stunting, temperature, and humidity) covariates	Multivariable logistic regression model	OR 3.00% [95% (CI): 0.54, 5.51%] for infant mortality

**Table 2 ijerph-20-03270-t002:** Summary of study designs and areas.

Characteristics		Frequency
Study area	LMIC [38,39,45]	3
	USA [22,41,42]	3
	South Korea [40,43]	2
	India [36,46]	2
	Africa [37,44,47]	3
Study design	Cohort [22,36,37,40,41]	5
	Cross-sectional [44,45,46,47]	4
	Case–control [42,43]	2
	Ecological [38,39]	2
Outcomes	Under-5 mortality [36,37,38,39]	4
	Post-neonatal mortality [22,40,42,43]	4
	Neonatal/post-neonatal/infant mortality [45,46]	2
	Infant mortality [41,44,47]	3
Cause of death	All-cause mortality [22,36,37,38,39,40,41,42,43,44,45,46,47]	13
	Respiratory mortality [22,37,40,42]	4
	Sudden infant death syndrome [22,40,42]	3

**Table 3 ijerph-20-03270-t003:** Summary of exposure assessment in selected articles.

Author and Year	Exposure Assessment	Source of PM2.5 and Constituents	Exposure Window	
			Post-Birth/Lifetime Exposure	In Utero/Prenatal and Post-Birth Exposure
Lien et al., 2019 [38]	Aerosol optical depth retrieved from MODIS	PM2.5 (biomass burning, anthropogenic pollutant, mineral dust, biomass/dust mixture, anthropogenic/dust mixture, and biomass/anthropogenic mixture)	■	
Owili et al., 2020 [39]	Aerosol optical depth retrieved from MODIS	PM2.5 (anthropogenic, biomass, and dust)	■	
Egondi et al., 2018 [37]	Ground-based monitoring station data	PM2.5	■	
Goyal et al., 2019 [45]	High-resolution calibrated satellite data	Dust and sea salt from other carbonaceous and manmade PM		○
Son et al., 2017 [22]	From PM2.5 prediction models based on satellite imagery	Ambient PM2.5	■	
Son et al., 2011 [40]	Ground-based monitoring station data	PM2.5 among other pollutants		○
Jung et al., 2020 [43]	CMAQ model	Ambient PM2.5		○
Woodruff et al., 2006 [42]	Ground-based monitoring station data	Ambient PM2.5	■	
Liao et al., 2022 [36]	Combination of monitoring station data, satellite-based AOD data, and random forest model	Ambient PM2.5		○
Heft-Neal et al., 2018 [44]	Data from satellite-based estimates of PM2.5	Ambient PM2.5		○
Khadka et al., 2021 [41]	Combination of ground-based monitoring and CMAQ model	Ambient PM2.5		○
deSouza et al., 2021 [46]	Satellite-derived PM2.5 concentrations	Ambient PM2.5		○
Bachwenkizi et al., 2021 [47]	Satellite-derived PM2.5 for mass and chemical transport model (GEOS-Chem)	Ambient PM2.5, including constituents such as organic matter (OM), black carbon (BC), sulfate, nitrate (NO3, NH4), and soil dust (DUST).	■	

■ Post-Birth/Lifetime Exposure, ○ In Utero/Prenatal and Post-Birth Exposure.

## Data Availability

Data sharing is not applicable, as no new data were created or analyzed in this study.

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
