# Peer review of "Association between Long-Term Ambient PM2.5 Exposure and under-5 Mortality: A Scoping Review"

_ijerph, 2023, doi:10.3390/ijerph20043270_

Round 1
Reviewer 1 Report
In general, this is a OK scoping review. However, I have three major concerns about the manuscript. These should be fixed before the manuscript can be considered again for publication:
The authors should define their exposure windows (e.g., prenatal, postnatal, long-term, etc.) more clearly in the earlier sections to avoid confusion. What do they mean, for instance, by "long-term"? How long is "long-term"? Is this comparable to post-natal? I have to keep going back and forth to figure out what they mean when they talk about multiple time windows in the article. It would be much easier for readers to follow along if, somewhere in the Methods section, the authors could insert a glossary table that defines these terms. Also, please don't use too many of these time terms, which can get very confusing very quickly as I read through the later parts of the manuscript.
Why was only post-natal exposure examined instead of both pre- and post-natal exposure? Is the relationship between prenatal PM 2.5 and mortality in children under five well-established? This does not appear to be the case (the authors also admitted this in the introduction). More justification should be provided for why the authors chose to focus on this limited exposure window scope (i.e., post-partum).
The English writing isn't great and can be very awkward at times. There are sentences that are very hard to understand and would take quite a bit of work to get the meaning. There are many mistakes in grammar throughout the article. If the authors want to be considered for publication again, they must undergo extensive editing of the English language and style.
Some detailed comments listed below:
Introduction:
1. According to the World Health Organization (WHO) almost the entire global population (99%) breath polluted air [5] and air pollution is one of the leading risks.
This is a really bold assertion. What constitutes air pollution? What year were the statistics collected? This is very ambiguous. The same questions apply to the sentence subsequent to this one.
2. What is the purpose of the second paragraph? Given that the result of interest is the infant mortality rate, Are these health issues related to morality rates?
3. I am not following the logic of the last two sentences in the third paragraph.
4. I do not follow the sentence that starts with "Elucidating ...
Materials and Methods
1. How about add "longitudinal" in the search term?
2. Have you identified additional articles after searching through the bibliographies of the identified articles? This is omitted in Figure 1.
Results
1. Section 3.4: Some of these articles may have examined covariates at the neighborhood level. For example, socioeconomic status, poverty rates, etc. In my experience, these are frequently included in environmental epidemiology research.
2. It would be advantageous to offer a summary of the statistical models employed to estimate effects, as the rigor of these models can impact the validity of the study's conclusions.
Discussion
1. I'm a little perplexed. The authors stated that they excluded prenatal exposure and only focused on postnatal exposure. However, the second paragraph continues to summarize prenatal outcomes? Or the scope of the investigation should be widened to incorporate additional exposure windows?
2. It appears that the second and third paragraphs cover very similar subject matter. Consider merging them to improve their flow. Currently, they read rather fragmentarily, with opinions scattered throughout.
3. Feel like a conclusion sentence is missing in the fourth paragraph.
4. Can you be more explicit here in terms of what do you mean by "Complexities relating to maternal health issues and infant mortality could explain these discrepancies."
5. The seventh paragraph: the authors frame this more like an issue that is more relevant to LMIC? So why would the high-income countries care about this topic? Or this is what the authors trying to argue? I am confused.
Author Response
We appreciate the very insightful comments from the reviewers and believe that these comments are extremely helpful to improve our manuscript. Here are the responses of the respective comments
Overall comments
In general, this is a OK scoping review. However, I have three major concerns about the manuscript. These should be fixed before the manuscript can be considered again for publication:
1.The authors should define their exposure windows (e.g., prenatal, postnatal, long-term, etc.) more clearly in the earlier sections to avoid confusion. What do they mean, for instance, by "long-term"? How long is "long-term"? Is this comparable to post-natal? I have to keep going back and forth to figure out what they mean when they talk about multiple time windows in the article. It would be much easier for readers to follow along if, somewhere in the Methods section, the authors could insert a glossary table that defines these terms. Also, please don't use too many of these time terms, which can get very confusing very quickly as I read through the later parts of the manuscript.
Author Response: We agree that these terms should be defined. The definitions of infant and child mortality is defined in section 2.3 of the method part. We inserted the following line to operationalize the term “long term” at the end of section 2.3 and rewritten the section as follows.
“Two reviewers extracted information on author name, publication year, study population, study location, study design, type of pollutant, duration of exposure, exposure measurement method, effect size (or findings), and health outcomes. Some of the studies analyzed the effect of prenatal or maternal exposure and lifetime postnatal exposure together to find the impact on infant mortality. Those studies defined the prenatal period as that spanning the beginning of the 1st trimester through the 3rd trimester of the gestational period. In the present study, we defined infant mortality as death within the first year of life (age, 0–364 days) and under-5 mortality as death of a child before they reached 5 years of age. Infant mortality was classified into two categories: neonatal mortality (age, 0–28 days) and post-neonatal mortality (age, 28–364 days). Effect estimates were presented as a percent change in excess risk with 95% confidence intervals (95% CIs) per 1 µg/m3 increase in ambient PM2.5. We defined long-term exposure as the post-birth lifetime exposure of the child from the day of their birth till the day of the outcome.”
- Why was only post-natal exposure examined instead of both pre- and post-natal exposure? Is the relationship between prenatal PM 2.5 and mortality in children under five well-established? This does not appear to be the case (the authors also admitted this in the introduction). More justification should be provided for why the authors chose to focus on this limited exposure window scope (i.e., post-partum).
Author Response: There have been a few studies on the association between prenatal exposure and infant mortality. However, the association between post birth exposure to ambient air pollution and under 5 mortality is less explored. Particularly, in developing countries, studies on air pollution and child mortality mainly focused on either indoor air pollution. To clarify this point, we have rewritten paragraph 3 of the Introduction part in the following manner.
“Exposure to particulate matter both before and after birth can increase the risk of infant and child mortality. Several studies estimated the risk of infant/child mortality from exposure to PM2.5 [15, 20-21], and some also estimated the risk for different exposure windows to explore the sensitive time period [15, 22-23]. These studies revealed that the association between exposure of ambient PM2.5 and infant mortality remains inconclusive. The association between post-birth ambient PM2.5 and under-5 mortality is also unclear as the impact of ambient air pollution on child mortality has not been quantified until recently. Many of these studies evaluated the association between indoor air pollution exposure and child mortality [24-26], focusing less on the association between long-term ambient PM2.5 and under-5 mortality. Moreover, determinants of neonatal mortality and under-5 mortality do not always overlap; this is because with neonates, maternal exposure to air pollution during the gestational period is often considered a key contributor to severe birth outcomes leading to neonatal death [27-29]”.
- The English writing isn't great and can be very awkward at times. There are sentences that are very hard to understand and would take quite a bit of work to get the meaning. There are many mistakes in grammar throughout the article. If the authors want to be considered for publication again, they must undergo extensive editing of the English language and style.
Author Response- We sent the article to the English language Editing Service to improve the language and amended the draft manuscript accordingly.
Some detailed comments listed below:
Introduction:
- According to the World Health Organization (WHO) almost the entire global population (99%) breath polluted air [5] and air pollution is one of the leading risks.This is a really bold assertion. What constitutes air pollution? What year were the statistics collected? This is very ambiguous. The same questions apply to the sentence subsequent to this one.
Author Response- For better understanding we have rephrased the corresponding lines in the following manner.
“According to the World Health Organization (WHO), air pollution is one of the leading risks to child health [5, 6]. In 2016, more than 90% of the world’s children, particularly the 630 million in LMICs, were exposed to higher levels of ambient particles with a diameter less than 2.5 μm (PM2.5) than those considered healthy by WHO guidelines [6]”
- What is the purpose of the second paragraph? Given that the result of interest is the infant mortality rate, Are these health issues related to morality rates?
Author Response: This paragraph explains vulnerability to PM2.5 for children. It also describes the previous evidence showing that PM2.5 increase the morbidity of various children’s diseases, which could increase. We added the following phrase to make it more logical.
“Previous epidemiological studies reported an association between exposure to PM2.5 and various health outcomes for children, such as respiratory infections [9-10], asthma [11-12], preterm birth [13-14], low birth weight [15], growth impairment [16], and neurodevelopmental [17-18] and metabolic [19] diseases, all of which can increase mortality risk.”
- I am not following the logic of the last two sentences in the third paragraph.
Author Response: We rephrased the sentences for clarification and also changed the corresponding reference 24 from the previous one:
“Moreover, the determinants of neonatal mortality and under 5 mortality is not always similar, because in case of neonates, maternal exposure to air pollution during the gestational period is often considered as a key factor for severe birth outcomes leading to neonatal death [27-29].”
- I do not follow the sentence that starts with "Elucidating ...
Author Response: We agree with that and therefore deleted the sentence
Materials and Methods
- How about add "longitudinal" in the search term?
Author Response: We agree that we did not add “longitudinal” in the search term at first. But after the comment from the esteemed reviewer we ran a search using the term” longitudinal” with our previous search terms. We have made changes in the section 2.1 in the method part as follows:
“We followed the guidelines of the PRISMA Extension for Scoping Reviews (PRISMA-ScR) [32]. A search strategy was developed and a literature search was conducted by combining search terms related to exposure (“ambient particulate matter” OR "outdoor air pollution” OR "outdoor PM2.5" OR “air pollutant”), health outcomes (“mortality” OR “death*”), target population ("child* OR "infant*" OR "under five"), time period of exposure (“long term”), and study design (“cohort studies” OR “cohort*” OR “epidemiological” OR “longitudinal”).”
We have also made changes in Figure 1 and in the corresponding part of the result section as follows:
“After screening by title and abstract, 400 articles were removed and the remaining 94 articles were included for full-text reading and assessed for eligibility. Among the 94 articles, 81 were excluded because they focused only on pregnancy outcomes (n=39), only presented results for adult populations (n=19), used different outcomes such as pregnancy outcomes or targeted other diseases such as anemia, cancer, and growth impairment (n=11). Non-English articles (n=7) and articles for which the full text was not available (n=2) were also excluded. We also excluded articles which used a definition of child that encompassed those aged >5 years (n=3). After the screen, a total of 13 articles remained for subsequent analyses (Table 1)”
Our search with the new term “longitudinal” did not change the number of included articles.
- Have you identified additional articles after searching through the bibliographies of the identified articles? This is omitted in Figure 1.
Author Response: We have not mentioned it in the figure before, but now edited it accordingly as follows:
“Studies selected for eligibility and full text screening including articles found from bibliography (n=94)”
Results
- Section 3.4: Some of these articles may have examined covariates at the neighborhood level. For example, socioeconomic status, poverty rates, etc. In my experience, these are frequently included in environmental epidemiology research.
Author Response: We appreciate this comment. We have rechecked the articles and revised section 3.4
- It would be advantageous to offer a summary of the statistical models employed to estimate effects, as the rigor of these models can impact the validity of the study's conclusions.
Author Response: We agree with this and we inserted a column for statistical methods employed for the included studies in Table 1 and also rewritten section 3.5 as follows-
“3.5.1 Post-birth exposure to PM2.5 and under-5 mortality-
Four studies discussed the effects of post-birth or lifetime exposure to ambient PM2.5 on under-5 mortality. In a study from India, Liao and colleagues used the Cox proportional hazard regression model and reported a 1.30% [95% (CI): 0.10%, 2.60%] increased risk of under-5 mortality for post-birth lifetime exposure to ambient PM2.5 and a 2.30% [95% (CI): 0.8%, 3.8%] elevated risk for in utero exposure for a 10 µg/m3 increase in ambient PM2.5 [40]. A study from Kenya using logistic and Poisson regression models reported that exposure to high levels of air pollution (PM2.5 ≥25 µg/m3) was associated with a 22.00% [95% (CI): 8.00%, 39.00%] increase in risk of mortality [44]. Meanwhile, Lien et al. reported a significant increase in risk of mortality for biomass PM2.5 exposure [29.00%; 95% (CI): 13.11%, 47.13%] and a moderate increase for anthropogenic PM2.5 exposure [12.00%; 95% (CI): 1.09%, 24.10%] by using a generalized additive mixed effects model [36]. Owili et al. reported an 8.90% [95% (CI): 4.11%, 13.91%] increase in risk of under-5 mortality for biomass PM2.5 and marginal or no associations with PM2.5 from other sources using the generalized linear mixed effects model [37].”
“3.5.2 Exposure to PM2.5 and infant mortality
The association between ambient PM2.5 exposure (both in utero and post-birth) and infant mortality was reported by 8 studies. Results for in utero exposure were dependent on trimester, cause of death, study design, and postnatal lifespan of the infant. Findings on the effects of in utero exposure in different trimesters on infant mortality were somewhat contradictory. Son et al., using the Cox proportional hazards model, reported a 14.50% [95% (CI): 6.70%, 22.80%] increased risk in infant mortality due to gestational exposure [46], and deSouza and colleagues, using fixed effect regression, reported an increase of 1.60% due to exposure only in the 3rd trimester [39]. Other studies found no association between gestational or in utero exposure with any stage of infant mortality [22, 38]. For postnatal or lifetime exposure, Heft-Neal et al., using the fixed-effects regression model, reported a 9.00% [95% (CI): 4.00%,14.00%)] increase in infant mortality for all causes for a 10 µg/m3 change in ambient PM2.5 [43], and Woodruff et al. reported a 7.00% [95% (CI): -7.33%, 23.55%] increase in all-cause mortality and a 113.00% [95% (CI): -12.01%, 305.04%] increase in deaths related to respiratory causes using the conditional logistic regression model [41]. For post-birth exposure, Son et al. used the extended Cox proportional hazards model and reported a 112.24% [95% (CI): 71.72%, 162.32%] increase in infant deaths in the United States [22], while Bachwenkizi and colleagues reported a 3.00% [95% (CI): 0.54, 5.51%] increase in infant deaths using multivariable logistic regression analysis [45]. In contrast, Khadka and colleagues reported no significant increase in infant mortality due to post-birth exposure to ambient PM2.5; notably, their study did not provide information about effect size [42]”
Discussion
- I'm a little perplexed. The authors stated that they excluded prenatal exposure and only focused on postnatal exposure. However, the second paragraph continues to summarize prenatal outcomes? Or the scope of the investigation should be widened to incorporate additional exposure windows?
Author Response: the 2nd paragraph in the discussion part is about the studies on under 5 mortality attributable to the postnatal exposure.
- It appears that the second and third paragraphs cover very similar subject matter. Consider merging them to improve their flow. Currently, they read rather fragmentarily, with opinions scattered throughout.
Author Response: We agree with the comment and we have merged and amended paragraph 2 and 3 as below.
“Among the selected studies, study designs differed mainly in studies targeting LMICs. The outcome for the selected studies was infant and child death attributable to long-term exposure to ambient PM2.5. The majority of studies suggested a positive association between lifetime exposure to ambient particulate matter (PM2.5) and infant and child mortality, although there was variation in effect size among studies, and even within the same study in some cases. Studies differed in design, sample size, exposure measurement method, and in many other aspects, which likely led to the observed variation. There were also specific differences between studies, such as the types and sources of PM2.5, which led to different effect estimates. For instance, Owili et al. reported that biomass PM2.5 was significantly associated with an increased risk of overall global under-5 mortality, whereas anthropogenic PM2.5 was not associated with under-5 mortality, although higher concentrations of anthropogenic PM2.5 were found in different parts of the study area [37]. Meanwhile, Lien et al. reported that both anthropogenic and biomass PM2.5 exposure elevated the risk of under-5 mortality overall in Asia [36]. For younger children, PM2.5 derived from carbonaceous and anthropogenic sources was associated with an increased risk of mortality [38]. The variation in results among the selected studies may be due to the different global frequency distributions of various PM2.5 types in different regions and time periods. In addition, the large study area and mixture of PM2.5 sources may have caused some overestimation or underestimation of results, as well as some inconsistencies. Therefore, it was difficult to gain a clear understanding of the impact of PM2.5 and its constituents on under-5 mortality”
- 3. Feel like a conclusion sentence is missing in the fourth paragraph.
Author Response: We agree and we added the following sentence at the end of the paragraph
“From these results, it was not possible to infer how post-birth exposure to PM2.5 influenced under-5 mortality.”
- 4. Can you be more explicit here in terms of what you mean by "Complexities relating to maternal health issues and infant mortality could explain these discrepancies?"
Author Response: We tried to mention the plausible reasons behind the variation in results for both infant and child mortality attributable to PM2.5 in the included studies. We have rewritten the line for more clarity as follows
“These discrepancies could be explained by the complexity of maternal health, as well as by our inadequate understanding of the biological mechanisms through which exposure to PM2.5 leads to infant mortality [47]”
- The seventh paragraph: the authors frame this more like an issue that is more relevant to LMIC? So why would the high-income countries care about this topic? Or this is what the authors trying to argue? I am confused.
Author Response: It has been reported in many studies that the health impact of PM2.5 is higher in LMICs due to several reasons including lack of resources to address both air pollution and the health burden. However several studies reported that exposure to low concentration of PM2.5 can elevate infant and child mortality and morbidity in high-income countries. In order to clarify the statement we added the following lines in the paragraph.
“However, other studies demonstrated that socioeconomic, racial, regional, and other disparities prevail in maternal health and infant/child mortality in high income countries [49], and that premature mortality can increase even in low concentrations of PM2.5 [50-51]”.

Reviewer 2 Report
Thank you for the opportunity to comment on your scoping review of the literature evaluating exposures to PM2.5 and mortality in children up to the age of 5 years.
The process you used to categorize and select relevant epidemiologic studies is logical and well-explained. As you describe, there is a wide range of study designs but very few that specifically examine exposure to PM2.5 and health outcomes in younger than school-age children. I agree that this may be a data gap.
From a public health perspective, there is no disagreement that exposure to PM2.5 in outdoor and indoor air along with associated pollutants is a hazard for adverse health effects regardless of age. Given that the primary causes of mortality in young children (infectious diseases, malnutrition, diarrhea, etc.) are frequently more prevalent in populations also exposed to high levels of PM (Sub-Sarahan Africa and some parts of South Asia), it is a challenge to untangle the effects of these concurrent exposures. Added to that is the challenge of characterizing exposure to PM2.5 since ambient air is a dynamic medium.
In your conclusion section, you state that more research is required because ambient air pollution is a major health risk and child mortality is still very high. As part of your future research, I hope you will consider the relative impacts of preventive health care interventions (i.e., vaccinations, nutrition, access to health care, etc.) compared to PM2.5 in terms of reducing childhood morbidity and mortality. Clearly reducing exposures to PM2.5 is an overarching goal, and we have made progress in some parts of the world. But as you point out, many of the areas that have high levels of ambient PM2.5 from natural or anthropogenic sources are also areas lacking in economic resources. I would suggest adding a short discussion to your conclusions section describing what you plan to do for the next steps in your research.
Author Response
Author Response: We thank you for your insightful comments and suggestions. We also agree that improved health care and relevant policy measures help in reducing the impact of air pollution on human health. Still more research is needed on the growing concern over global climate change and the joint impact of climate change and air pollution on human health. According to your suggestion we amended the conclusion part as follows:
“The present scoping review highlighted the lack of evidence in the literature on the effects of long-term exposure to ambient PM2.5 on child mortality. More research is required in this field since ambient air pollution is a major public health risk and, in some parts of the world, child mortality is still very high. The global community is committed to achieving SDG targets by 2030, and Goal 3 of this global blueprint aims to reduce both under-5 mortality and the number of deaths and illnesses from environmental pollutions. The tracking record of SDG progress shows that many countries–especially LMICs—will not be able to meet the targets by 2030 due to a number of reasons, research and data gaps being two of the most glaring to date. Therefore, in order to implement necessary policy measures to avoid excess mortality and achieve SDG targets on time, it is exclusively important for governments, academia, and all other stakeholders to understand the pertaining risk of excess death among under-5 children due to exposure to long-term ambient PM2.5 pollution. Environmental health research can focus on obtaining more epidemiological evidence to understand the current association between PM2.5 and under-5 mortality, and how population dynamics and interventions such as improved health care facilities and air pollution control measures can change the burden of mortality. The exposure response relationship ascertained from these studies could be applied to projection studies for under-5 mortality in the coming years, especially amidst the growing concerns about the impact of global climate change.”

Round 2
Reviewer 1 Report
The authors have adequately addressed my questions and concerns. I have no further comments.